# Virtual Scoliosis Surgery Using a 3D-Printed Model Based on Biplanar Radiographs

**DOI:** 10.3390/bioengineering9090469

**Published:** 2022-09-14

**Authors:** Aurélien Courvoisier, Antonio Cebrian, Julien Simon, Pascal Désauté, Benjamin Aubert, Célia Amabile, Lucie Thiébaut

**Affiliations:** 1TIMC, University Grenoble Alpes, CNRS, UMR 5525, VetAgro Sup, Grenoble INP, CHU Grenoble Alpes, 38000 Grenoble, France; 2Grenoble Alps Scoliosis and Spine Center, Grenoble Alps University Hospital, Bvd de la Chantourne, 38043 Grenoble, France; 3EOS Imaging, 10 rue Mercoeur, 75011 Paris, France; 4eCential Robotics, Zone Mayencin II, Parc Equation-Bât.1, 2 Avenue de Vignate, 38610 Gières, France

**Keywords:** 3D printing, bi-planar X-rays, virtual surgery, scoliosis, additive manufacturing

## Abstract

The aim of this paper is to describe a protocol that simulates the spinal surgery undergone by adolescents with idiopathic scoliosis (AIS) by using a 3D-printed spine model. Patients with AIS underwent pre- and postoperative bi-planar low-dose X-rays from which a numerical 3D model of their spine was generated. The preoperative numerical spine model was subsequently 3D printed to virtually reproduce the spine surgery. Special consideration was given to the printing materials for the 3D-printed elements in order to reflect the radiopaque and mechanical properties of typical bones most accurately. Two patients with AIS were recruited and operated. During the virtual surgery, both pre- and postoperative images of the 3D-printed spine model were acquired. The proposed 3D-printing workflow used to create a realistic 3D-printed spine suitable for virtual surgery appears to be feasible and reliable. This method could be used for virtual-reality scoliosis surgery training incorporating 3D-printed models, and to test surgical instruments and implants.

## 1. Introduction

The global 3D-printing market is growing in the field of orthopedics and spine surgery [1,2,3]. Companies and surgeons are increasingly adopting 3D printing for custom or patient-specific implants [4,5,6,7,8]. 3D printing is also becoming ever more popular for surgical planning [3,9,10,11,12,13,14,15] as well as for educational purposes.

Recent studies have highlighted that 3D-printed spines seem feasible, cost effective [16], and may enhance the education of trainees [17,18,19,20]. Several authors have showed that 3D ex vivo modeling of case-specific physiologic and pathologic conditions can improve the effectiveness of medical education due to the strong influence of the 3D effect [21,22,23,24,25,26]. Burkhard et al. [27] confirmed that a 3D-printed vertebral column displayed a haptically and biomechanically realistic simulation for posterior spinal procedures, notably for screw positioning [22]. In addition, the use of 3D-printed models has been reported to improve patient–surgeon communication, thus potentially expanding patient compliance and decreasing patient anxiety [21]. The apparent benefits and far-reaching applications of this growing technology make it an attractive option for the future of spine surgery [8]. 

Operations for spinal deformity are some of the most complex procedures performed in pediatric orthopedics [28], and can result in prolonged operative times and significant blood loss. Additionally, there is a major learning curve for the placement of thoracic pedicle screws in patients with AIS [29]. Posterior fusion for AIS is, of course, not only a matter of screw placement: education and acquisition of all the other aspects of the operation are also essential to improve the quality of care, patient safety, and outcomes. The applications of 3D-printed models have to date been restricted to simple surgical procedures [18,20,23,24,30]. We hypothesized that reproducing the entire operation on a patient-specific 3D-printed scoliotic spine would improve the training and future performance of surgeons treating patients with complex spinal deformities.

The majority of 3D-printed spine models reported in the literature have been constructed from CT or MRI images [3,18]. Although CT scanning is rapid and generally more accessible, the patient receives a significant radiation dose. If the standard preoperative work-up does not include a CT scan, acquiring such images for the sole purpose of 3D printing would be difficult to justify ethically in terms of the radiation exposure involved. 

An alternative imaging technique is the EOS system (EOS imaging, Paris, France), which provides full-body bi-planar low-dose standing X-rays and can generate a 3D spine reconstruction [31]. Since EOS images are already integrated into the standard care package for AIS patients in our center, using this technique to obtain individualized 3D spine geometry with limited radiation exposure is a logical solution for 3D printing [32].

This study aims to describe a protocol that replicates the actual spinal surgery undergone by patients with AIS by using a 3D-printed “twin” spine model based on biplanar radiographs to perform virtual surgery. In this study, two patients with AIS were recruited and imaged pre- and post-operatively, allowing a comparison between actual surgical outcome and the virtual surgical outcome utilizing the 3D-printed scoliotic spine. 

## 2. Materials and Methods

### 2.1. Patients and Surgical Technique

Two adolescent patients presenting with severe Lenke 1A idiopathic scoliosis were recruited. Their clinical and radiological features are presented in Table 1. Standard pre- and postoperative EOS spinal imaging was undertaken. Posterior spinal fusion was assisted by the Surgivisio^®^ (eCential Robotics—Gières–France) navigation system [33] based on cone-beam computed tomography (CBCT). 

Posterior spinal fusion was performed in both cases using hybrid constructs (hooks and screws at the upper extremity of the construct, sublaminar bands at the apex of the curve, and pedicular screws at the lower end of the construct). Screw placement was confirmed using the Surgivisio system.

### 2.2. Conceptual Model

The concept behind the study presented in this paper is to replicate the main steps of the care pathway for a given surgical pathology (in this case, AIS) in order to evaluate the technical solutions that could improve each step of treatment. By producing a realistic setup with a simple method, we anticipate accelerating the evaluation and refinement of new medical devices, a validation that would otherwise normally require complex and lengthy clinical trials. 

The process presented in Figure 1 was established to replicate a typical AIS surgical care pathway. A control loop involving a comparison between the real patient and their artificial 3D-printed twin was included at each main step to verify the relevance of the artificial twin.

### 2.3. Imaging for Subject-Specific 3D Modeling

Spinal 3D modeling acquired from the pre- and post-operative bi-planar X-rays was achieved using sterEOS (sterEOS_1.8.7.66R-EOS imaging—Paris-France) according to the method described by Humbert [31]. The following spinopelvic parameters were calculated pre- and postoperatively: T1T12 thoracic kyphosis (TK), L1S1 lumbar lordosis (LL), Cobb angle(s) and axial rotation of the apical (most tilted) vertebra, pelvic incidence (PI), sacral slope (SS) and pelvic tilt (PT).

In order to obtain a better estimation of the 3D shape of the spine intra-operatively (e.g., patient in a prone position), the preoperative 3D reconstructions in the standing position were matched to the CBCT image volume. Matching was performed by (1) manual rigid registration of the 3D models extracted from sterEOS onto the CBCT images and (2) fine-tuning of the shape by adjusting the 3D segmentation on all slices using the segmentation tools in the 3D Slicer open source software (https://www.slicer.org, accessed date: 9 July 2022) [34]. The final re-adjusted 3D shape of the spine in the prone position was used to print the individual 3D models of each vertebrae. A more simplified 3D printing workflow could be envisaged by merely using the 3D sterEOS model in the standing position. This is likely to be sufficient for replicating the surgery for training purposes. In this current study, the aim was to test the model possessing the most realistic accuracy. 

### 2.4. 3D Printing of the Spine

The spine was split into three distinct types of structures (vertebrae, shell to link the bones, and a support structure) to obtain realistic 3D-printed models that would be adequate for the entire length of the surgical operation. This choice of architecture was driven by the following constraints:To enable standing and prone positions of the 3D-printed spineTo reproduce the mechanical behavior of the actual patient spineTo be compatible with X-ray-based imaging techniques

#### 2.4.1. Vertebrae

To model the vertebrae in a realistic manner, special consideration was given to the materials used for 3D printing. Firstly, a study was conducted to identify materials offering an appropriate radio-opacity. As the concept was to simulate the classical surgical care pathway followed by a patient with AIS, the 3D spine models needed to be imaged pre- and postoperatively, as well as potentially intra-operatively by a navigation system, and offer similar X-ray images to those of the actual patient. To reproduce the radiological appearance of the vertebrae, various materials were tested on 1-mm-thick sample plates (grey scale for each sample can be seen in Figure 2):Polylactic acid was too too opaque.Polylactic acid filled with 80% copper was transparent.Polylactic acid with 50% ceramic was identified to reproduce the radio-opacity of bone most closely.

Having selected the appropriate materials, the 3D printer settings were evaluated, in particular concerning the grey scales for the representation of cancellous bone (comparable to that of an actual vertebral X-ray). The following 3D printing settings were tested, employing a printing density between 20 and 30%:Filling “quarter cubic 45°” (pre-set in the 3D printer);Filling “hole’s infill” (pre-set in the 3D printer);Filling “personalized vertical grid 5°” (customized).

Three vertebrae (each 3D-printed with one of the above settings) were then imaged with EOSedge^®^ (the latest EOS imaging device). Figure 3 shows that the customized filling “vertical grid 5°” setting obtains the most realistic radiographic rendering, with a uniform texture of cancellous bone and a significant signal in cortical bone. Bi-planar X-ray images of actual patient vertebrae and the surrounding soft tissues are presented in Figure 4 for comparison.

#### 2.4.2. Shell

To maintain the static alignment of the vertebrae while simultaneously allowing some displacement during fixation of the spinal rods, a shell linking the vertebrae together was created to mimic the intervertebral disks and anterior and posterior longitudinal ligaments. This shell, 3D printed in thermoplastic polyurethane, covers all the vertebral bodies (Figure 5) except for the pedicles and the central aspect of the posterior longitudinal ligament (space allowing for the insertion and removal of the vertebrae).

#### 2.4.3. Pelvis and Support Structure

A support structure was created by simplifying the pelvic 3D model to retain only the areas of interest. The acetabular cups and sacral plate were connected by shafts and positioned on a support that provides stability for the entire structure. An accessory part in the form of an intervertebral disc between the sacrum and the L5 vertebra was designed to receive the vertebral column. Studs on the upper side of the disc allow for reception of the spine assembled in the shell. Drill holes are provided in the 3D-printed L5 vertebra as well as in the lower face of the shell (Figure 6). From this point onwards in the paper, “3D-printed twin” refers to the totality of the following elements: spine, shell, pelvis, and support structure.

The simplified pelvis and support were removed during the virtual surgery and were only used to acquire both pre- and postoperative bi-planar X-rays. To simulate the deformation of the anterior soft tissues during surgery, a dedicated foam was used (conforming to the anterior curve of the 3D-printed twin) to support the 3D-printed spine during surgery (Figure 5).

### 2.5. 3D-Printed Models’ Assessment

Adequacy of the mechanical behavior of the 3D-printed twin during pedicle screw positioning was then evaluated. As a first step, the mechanical integrity of an intervertebral unit composed of three 3D-printed vertebrae was assessed during screw drilling. A vertebra with inserted pedicle screws can be seen in the picture and the bi-planar X-rays in Figure 7.

Initial analysis revealed a vertebral structure possessing excessive fragility in the plane perpendicular to the direction of impression. This directional weakness was resolved by adding an epoxy resin coating at the terminal stage of vertebrae fabrication, which, in addition, improved the realistic mechanical behavior of the cortical bone. 

Secondly, the overall mechanical behavior of the spine was also assessed within an intervertebral unit consisting of seven vertebral levels. The resulting analysis engendered several adjustments of the cuts made in the shell around the posterior ligament.

### 2.6. Virtual Surgical Procedure on 3D-Printed Models

#### 2.6.1. Preoperative and Postoperative Imaging

During the pre- and postoperative stages of the virtual surgical procedure, the 3D-printed twin was subjected to the same protocol as the real patient: standing bi-planar X-ray imaging using the EOS system. To enable image acquisition, full spinal support was implemented to maintain the standing position of the spine. Surgical planning was then undertaken based on the preoperative images and the dedicated EOS software.

#### 2.6.2. Virtual Surgery

The setup for the virtual surgery is illustrated in Figure 5. The standing support of the 3D-printed twin was replaced by the foam support on which the 3D-printed spine was positioned in a prone position.

As the number of implants and tools available were limited in the demonstration sets during the virtual surgery on the printed models, it was not possible to perfectly duplicate the bony constructs performed on the actual patients. Nevertheless, the length of the constructs, using similar upper and lower instrumented vertebrae, was respected. Only hooks and screws were used in the virtual operations due to the unavailability of sublaminar bands. The absence of a rod cutter for the procedure on the first twin explains the overlength of the two rods seen in Figure 8.

Once the 3D CBCT images of the levels to be instrumented were acquired with the Surgivisio imaging platform [35], screws were placed using the associated navigation system. Rods were bent to fit the correct expected sagittal balance. The concave rod was the first to be inserted. Correction maneuvers were then applied as follows: rotation of the rod, then contraction between the implants at the upper end of the construct to stabilize the claw between the supralaminar hook and the screws, and screw distraction at the concave side. The convex rod was inserted to neutralize the construct. Similar maneuvers were applied at the upper end, but on this side, contraction between the lower screws was performed. 

## 3. Results

### 3.1. 3D Printing

The pre- and postoperative constructs of one of the 3D-printed spine twins can be seen in Figure 9. Figure 8 (Patient 1) and Figure 10 (Patient 2) demonstrate pre- and postoperative bi-planar X-rays of the actual patient and their 3D-printed twin spine.

During surgery on the 3D-printed twin 1, pedicles were damaged and fractured during pedicle screw positioning and drilling. Although this can occur during real surgery, the decision was made to reinforce the vertebrae of 3-D printed twin 1 with an epoxy resin coating. No damage was observed during screw placement on the 3D-printed twin 2.

### 3.2. Spinopelvic Parameters of Both Pairs of Patients and 3D-Printed Twins

Characteristics and spinopelvic parameters of both patients and their 3D-printed twins can be found in Table 2. The twin was printed based on the form of the patient in the prone position during the surgery. In the prone position, the scoliosis curvature and sagittal parameters decrease, which explains the difference between the main preoperative curve (Preop Curve 2) between both patients and their twins, and also the difference between patient 1 and twin 1 regarding preoperative lordosis (Preop Lordosis). The degree of preoperative lordosis in patient 2 was small; thus, only a slight difference was found between the patient and their twin. 

The difference in the correction of the main curve between patient 2 (16°) and twin 2 (27°) is due to a lack of instrumentation at the apex of the curve in the twin. In patient 2, several sublaminar bands were used to optimize the correction.

### 3.3. Surgeon Feedback from Virtual Surgery

The 3D-printed twin provides a “feeling” for screw placement similar to the sensation experienced during real surgery. The improvement in cortical stiffness obtained with the epoxy coating allowed proper screw fixation for the second twin, whereas screw placements for the first twin (without the vertebral coating) tended to split the pedicle.

Hook positioning relies on the preparation of the soft tissues in and surrounding the intervertebral space between the laminae (yellow ligament). Due to the absence of soft tissues within the posterior region of the 3D-printed spine, hook placement was rendered effortless, and thus unrealistic. In our practice, hybrid constructs are preferred to all-screw constructs. However, the sublaminar bands that play an integral role in the correction were unavailable for the purpose of this study.

Once the anchors were in place, rod contouring and placement produced similar corrections of the 3D spine as those achieved in the real surgery. The principal issue in the final stages of the virtual surgery was that the foam support serving to position the spine twin during the surgery proved less flexible than he twin. Upon obtaining spinal correction, the twin could no longer fit into the foam support.

## 4. Discussion and Conclusions

This study presents a method to configure individualized 3D-printed spines using 3D geometrical features obtained from preoperative X-rays of adolescents with idiopathic scoliosis. The 3D-printed “twin” spines are then used to simulate the spinal fusion surgery undergone by AIS patients. Two AIS patients planned for surgery were recruited for the purpose of the study.

In terms of the postoperative clinical parameters, 3D-printed twins were close enough (within 10°) to the real postoperative 3D patient spines. Similar corrections were achieved for both patients and their matching 3D-printed twins. Lack of a number of instruments and anchors during the virtual surgeries explains the slight difference in the patient/twin n°2 Cobb angle of curve 2 (16° vs. 27°). On the contrary, the sagittal parameters were similar. This is not surprising since the sagittal parameters (LL and TK) rely on rod contouring, which was performed in the twins according to the patient’s actual postoperative X-rays. Despite some differences in the constructs and the operating conditions, the changes in clinical parameters following surgery were similar for the patients and their 3D-printed twins, testimony as to the realistic nature of the virtual surgery.

The optional adjustment of the 3D sterEOS model to better fit the prone shape of the spine intraoperatively explains the radiological differences between the preoperative Cobb angle of the patient and the twin (61° vs. 47° for pair 1 and 53° vs. 27° for pair 2). As explained earlier, this step is optional; the 3D sterEOS model could have been sufficient to simulate the surgery. The differences between the 3D EOS model and the CBCT prone position adjusted model were observed on the Cobb angle and on the LL (10° to 20° difference). The TK was less affected (5° to 10° difference). The preciseness of the 3D-printed twin described in the current study resulted in a highly realistic model; however, the greater complexity of the construction increases manpower, time, and costs.

Evaluation of the difference of the spine shape between the standing and prone positions confirms the recognized fact that coronal and sagittal parameters are altered in the prone position. Since the prone position may vary from one operating room table to another, it is difficult to generate a standardized prone position. Additionally, the magnitude of these variations also depends on spine flexibility, which is patient-specific. This is the reason why it is optional to take such variations into account for 3D printing a spine model for routine training or demonstration purposes.

While 3D-printing applications in spine surgery are gaining momentum, some challenges remain to be addressed. In particular, realistic haptic simulation of the 3D-printed model needs to be sufficiently reliable for the surgeon [22,27]. Another challenge reported is the workflow and time associated with 3D printing due to time-consuming 3D volume segmentation from CT or MRI images, as well as the access and availability of 3D printers. These constraints may have limited the embrace of this technology up till now. Costs associated with 3D printing can be prohibitive, although, it appears possible to drastically reduce associated costs if work volume is increased [20]. Finally, relying on CT images for 3D spine modeling is a limitation due to the radiation dose involved in imaging the whole spine.

In this study, most of these issues were able to be circumvented. The patient-specific 3D-printed models were configured from routine full-spine EOS bi-planar radiographs at a reasonable cost and low radiation exposure. The estimated cost of the 3D-printed twin 2 (with epoxy resin coating to reinforce spine integrity during surgery) is around 1000 Euros, which is on par with other reported price estimations. As the cost associated with the 3D-printing workflow is volume-based, cost would be expected to be lowered if a higher volume of 3D-printed models of the same spine is demanded.

We simulated realistic bone tissue from both radiological and surgical perspectives. Our 3D-printed models were constructed by plugging each printed vertebrae in a soft shell, simulating the intervertebral discs and surrounding ligaments. It would be possible to simulate different types of shells, varying from highly flexible to very rigid. Similarly, one could potentially generate diverse types of bone quality to test the behavior of implants. The main limitation in the model developed is the absence of the surrounding truncal soft tissues. Ideally, the 3D twin should have an integrated rib cage with a parametric support mimicking the mechanical behavior of the soft tissues. Work is in progress to generate a parametric 3D twin in which soft tissue elasticity and bone density can be monitored. The next step of our work is to generate all of the patient-specific trunk anatomy, which entails including the ribs, chest wall, and muscles. These elements should radically improve the realistic nature of the 3D twin. Our opinion is that preoperative simulation and/or reperforming surgery on 3D-printed twins will improve both the efficiency and safety of spinal operations, particularly for junior surgeons and surgeons wanting to accelerate their learning curve for new techniques and implants.

The proposed process widens the perspective of enhancing surgical training for the correction of spinal deformities, particularly at the beginning of training and for severe cases. It could also be implemented to test or demonstrate the utilization of new instruments and implants.

## Figures and Tables

**Figure 1 bioengineering-09-00469-f001:**
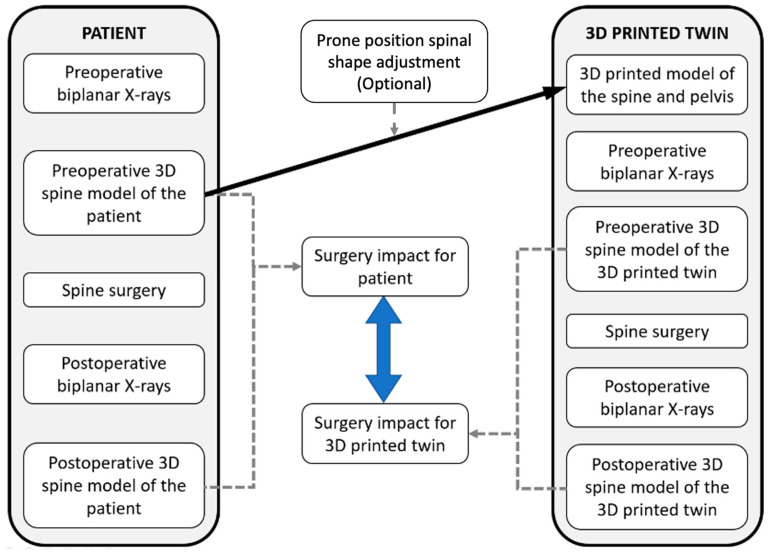
Process for each patient.

**Figure 2 bioengineering-09-00469-f002:**
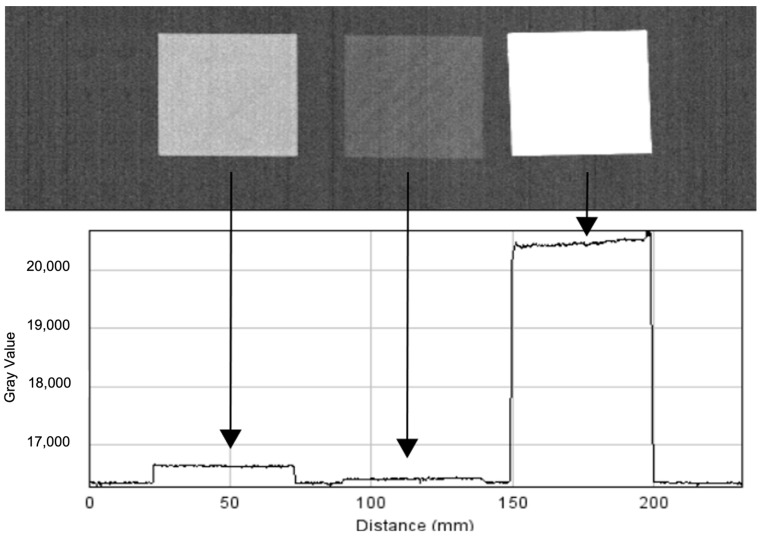
Grey levels of 1-mm-thick sample plates with different materials: polyactic acid with 50% ceramic (**left**), polyactic acid filled with 80% copper (**middle**), and polyactic acid (**right**).

**Figure 3 bioengineering-09-00469-f003:**
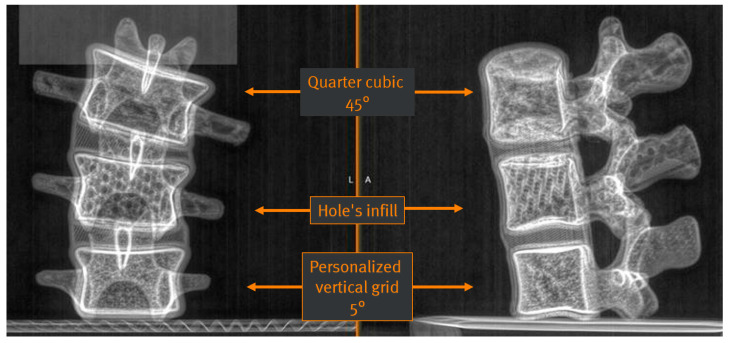
Bi-planar X-rays of three 3D-printed vertebrae, each resulting from different printer settings. From top to bottom: quarter cubic 45°, hole’s infill, and personalized vertical grid 5°.

**Figure 4 bioengineering-09-00469-f004:**
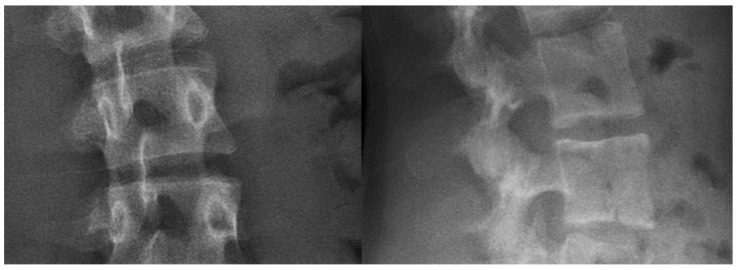
Bi-planar X-rays of the spine of an actual patient.

**Figure 5 bioengineering-09-00469-f005:**
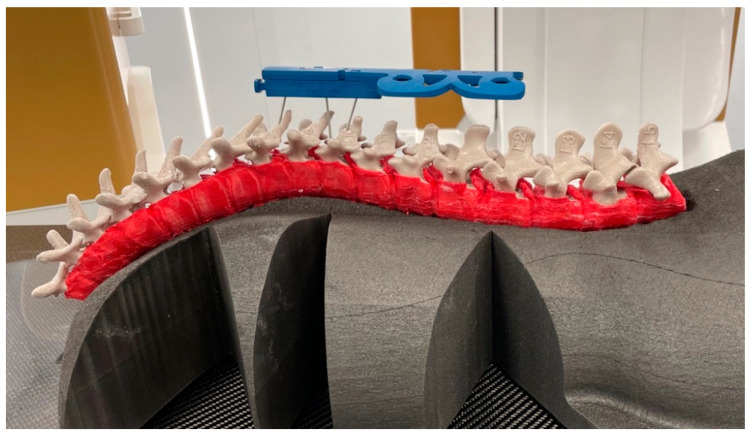
3D-printed shell (red) linking the vertebrae and positioned for surgery on a specifically designed foam.

**Figure 6 bioengineering-09-00469-f006:**
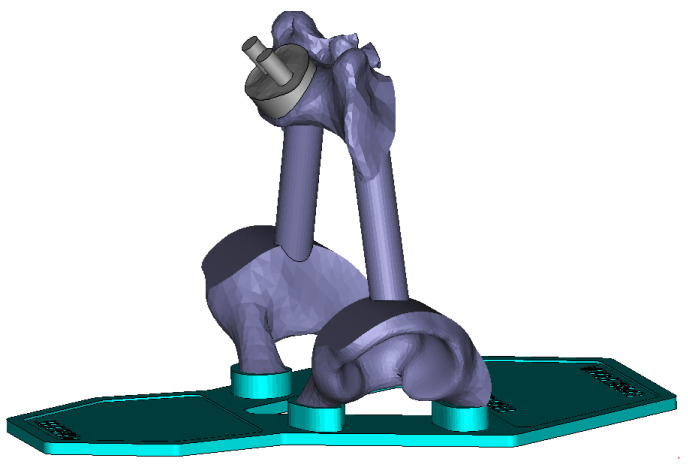
3D-printed simplified pelvis and support structure.

**Figure 7 bioengineering-09-00469-f007:**
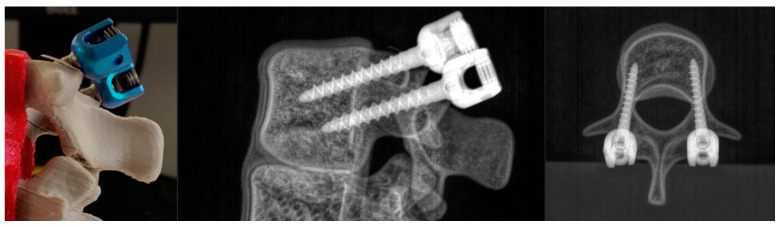
Picture and oblique and superior X-rays of a 3D-printed vertebra with two pedicle screws inserted.

**Figure 8 bioengineering-09-00469-f008:**
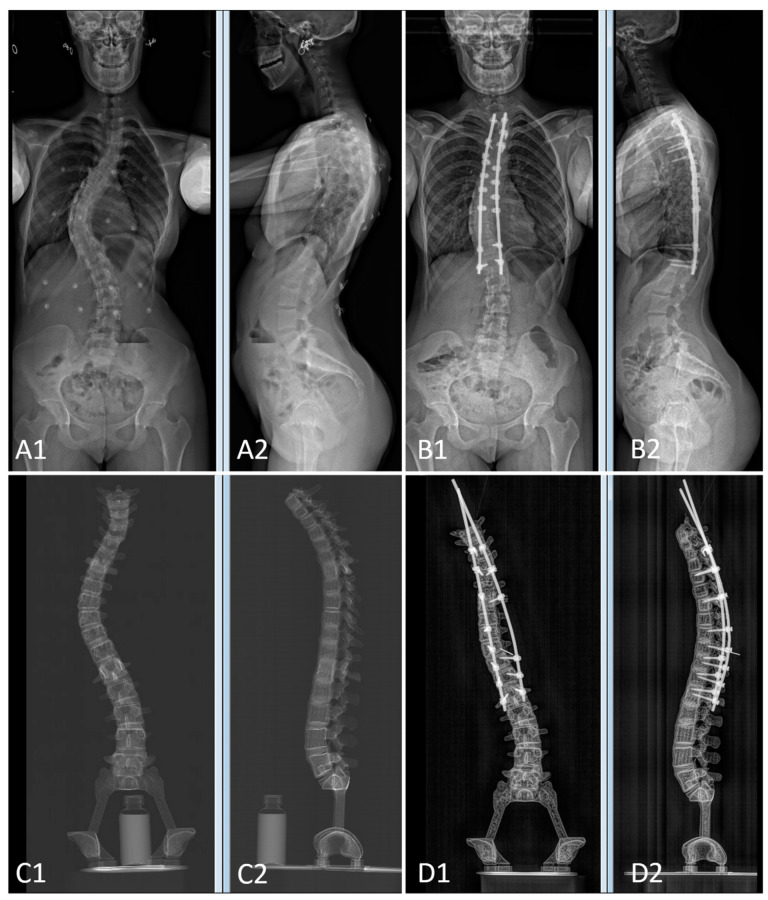
Bi-planar X-rays of patient 1 and 3D-printed twin 1. (**A**): Preoperative X-rays of patient 1 ((**A1**): frontal view, (**A2**): lateral view). (**B**): Postoperative X-rays of patient 1 ((**B1**): frontal view, (**B2**): lateral view). (**C**): Preoperative X-rays of 3D-printed twin 1 ((**C1**): frontal view, (**C2**): lateral view). (**D**): Postoperative X-rays of 3D-printed twin 1 ((**D1**): frontal view, (**D2**): lateral view).

**Figure 9 bioengineering-09-00469-f009:**
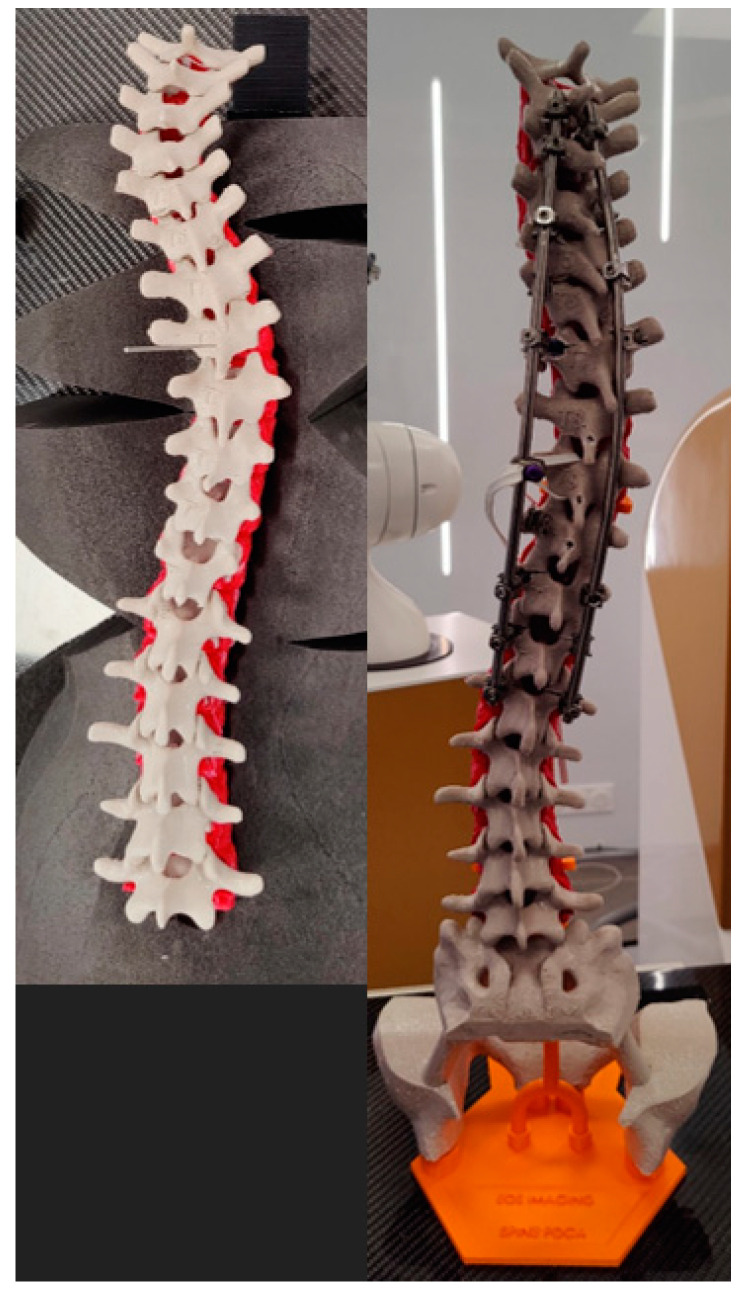
Pictures of posterior view of 3D-printed twin 2. On the left, preoperative 3D-printed twin in prone position without pelvis and support structure. On the right, 3D-printed twin after surgery in a standing position.

**Figure 10 bioengineering-09-00469-f010:**
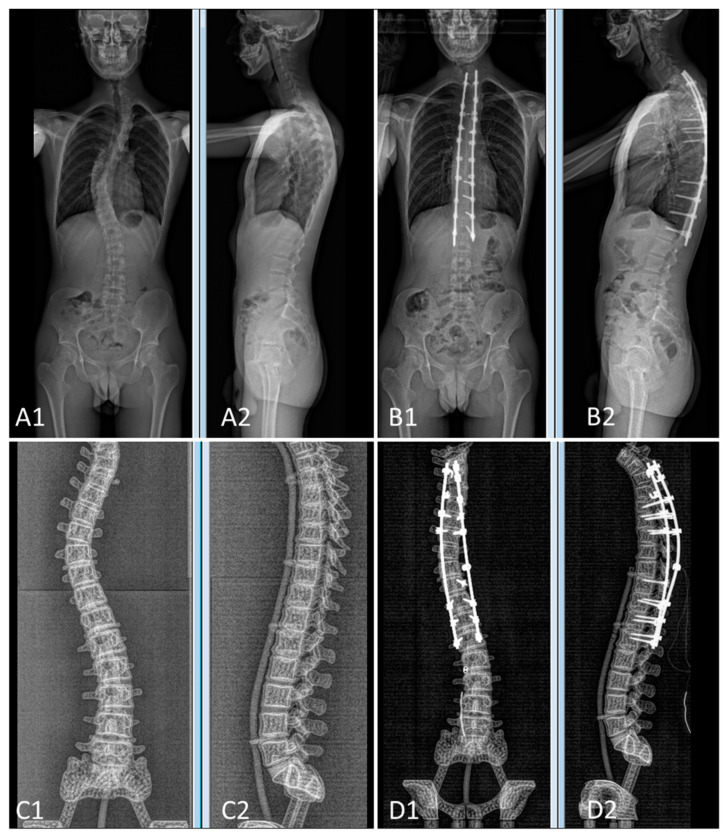
Bi-planar X-rays of patient 2 and 3D-printed twin 2. (**A**): Preoperative X-rays of patient 2 ((**A1**): frontal view, (**A2**): lateral view). (**B**): Postoperative X-rays of patient 2 ((**B1**): frontal view, (**B2**): lateral view). (**C**): Preoperative X-rays of 3D-printed twin 2 ((**C1**): frontal view, (**C2**): lateral view). (**D**): Postoperative X-rays of 3D-printed twin 2 ((**D1**): frontal view, (**D2**): lateral view).

**Table 1 bioengineering-09-00469-t001:** Cohort description and spinopelvic parameters.

Patient		Patient 1			Patient 2		
**Age (years)**			14			18	
**BMI (kg/m^2^)**			26			24	
**Levels instrumented**		T4L1			T2L1		
**Exam**		Preop	Postop	Change	Preop	Postop	Change
**Pelvic parameters**	**PI (°)**	39	40	1	53	52	−1
	**PT (°)**	−9	−8	2	14	16	2
	**SS (°)**	49	48	−1	39	36	−3
**Sagittal curvatures**	**TK T1T12 (°)**	41	40	0	40	55	16
	**LL L1S1 (°)**	73	55	−19	55	53	−2
**Coronal curve 1**	**Levels**	L1-L3-L4	L1-L3-L4	NA	T12-L3-L5	T12-L3-L5	NA
	**Cobb (°)**	32	27	−5	22	5	−17
**Coronal curve 2**	**Levels**	T7-T10-L1	T7-T10-L1	NA	T6-T9-T12	T6-T9-T12	NA
	**Cobb (°)**	61	32	−29	53	16	−37
**Coronal curve 3**	**Levels**	T1-T3-T7	T1-T3-T7	NA	T1-T3-T6	T1-T3-T6	NA
	**Cobb (°)**	40	29	−11	44	27	−17

**Table 2 bioengineering-09-00469-t002:** Spinopelvic parameters and curves for the 3D-printed twins (corresponding parameters for the actual patients are presented in smaller font within brackets).

Radiological Parameters		3D-Printed Twin 1			3D-Printed Twin 2		
		Preop	Postop	Change	Preop	Postop	Change
**Pelvic parameters**	**PI (°)**	40 (39)	40 (40)	0 (1)	54 (53)	57 (52)	3 (−1)
	**PT (°)**	−6 (−9)	−8 (−8)	−2 (2)	13 (14)	15 (16)	2 (2)
	**SS (°)**	46 (49)	48 (48)	2 (−1)	41 (39)	42 (36)	1 (−3)
**Sagittal curvatures**	**TK T1-T12 (°)**	46 (41)	40 (40)	−6 (0)	51 (40)	59 (55)	8 (16)
	**LL L1S1 (°)**	53 (73)	61 (55)	8 (−19)	45 (55)	53 (53)	8 (−2)
**Coronal curve 1**	**Levels**	L1-L3-L4	L1-L3-L4	Na	T12-L3-L5	T12-L3-L5	Na
	**Cobb (°)**	29 (32)	29 (27)	0 (−5)	23 (22)	22 (5)	−1 (−17)
**Coronal curve 2**	**Levels**	T7-T10-L1	T7-T10-L1	Na	T6-T9-T12	T6-T9-T12	Na
	**Cobb (°)**	47 (61)	27 (32)	−20 (−29)	37 (53)	27 (16)	−10 (−37)
**Coronal curve 3**	**Levels**	T1-T3-T7	T1-T3-T7	Na	T1-T3-T6	T1-T3-T6	Na
	**Cobb (°)**	19 (40)	28 (29)	9 (−11)	37 (44)	23 (27)	−14 (−17)

## Data Availability

Not applicable.

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
