# Peer review of "Virtual Scoliosis Surgery Using a 3D-Printed Model Based on Biplanar Radiographs"

_bioengineering, 2022, doi:10.3390/bioengineering9090469_

Round 1
Reviewer 1 Report
The authors described a method to simulate spine surgery for adolescents with idiopathic scoliosis using a 3D-printed model of their spine in the article. Using low radiation full spine x-rays, the researchers demonstrated that it's possible to create realistic, patient-specific 3D printed models that are affordable. This was one of the main reasons these papers are noteworthy. Methods like the one proposed here can be used to train fellows with different levels of experience, as well as to demonstrate the use of new instruments or implants. It is novel enough for me to recommend this article to be published in its current state.
Author Response
Thanks a lot for your comments.
The manuscript has underwent corrections and been resubmitted according to the other reviewer comments.
Reviewer 2 Report
This study describes using the artificial 3D-printed model to represent the spines of adolescents who suffered from AIS. Simulation regarding spine surgery was conducted with a global workflow for each patient produced. The topic is interesting and realistically valuable. However, several flaws exist and they must be addressed carefully.
a. current Introduction is not sufficient enough to give readers a whole and clear picture of what authors intent to do, and what have been achieved in this study. Aim, process, and result need to be re-structured to ensure a better logic flow. Also, more background knowledge about the aspects that are mentioned in this paper need to be briefly/moderately introduced, possibly with references. This helps improve the scientific soundness.
b. The global workflow actually is a contribution from this study. But there lacks justification on how this workflow is produced, for example, it is based on the experiment/simulation result, or it was already there before initiating the experiment. Therefore more detailing discussion is needed.
c. It seems like there are still more works for the authors' project team to continue, after fabricating and simulating the artificial spines. For future works and recommendations on improving current work, there needs more discussion.
Other suggestions:
1. For the 'materials' selection, three samples were compared, the last is best but there is no reason why the last one is optimal to reproduce realistic radiolucency of the vertebrae.
2. There are many linguistic errors and slips, as well as the reference errors via translating Words into PDF. Grammar and long sentences throughout the whole paper need to be checked and improved.
In brief, the value of the focused topic in this paper is high, but several aspects as mentioned, as well as the general readability, need to be addressed at first place.
Author Response
a. current Introduction is not sufficient enough to give readers a whole and clear picture of what authors intent to do, and what have been achieved in this study. Aim, process, and result need to be re-structured to ensure a better logic flow. Also, more background knowledge about the aspects that are mentioned in this paper need to be briefly/moderately introduced, possibly with references. This helps improve the scientific soundness.
R: Introduction has been improved.
b. The global workflow actually is a contribution from this study. But there lacks justification on how this workflow is produced, for example, it is based on the experiment/simulation result, or it was already there before initiating the experiment. Therefore more detailing discussion is needed.
R: The workflow was justified and detailed
c. It seems like there are still more works for the authors' project team to continue, after fabricating and simulating the artificial spines. For future works and recommendations on improving current work, there needs more discussion.
R: Perspectives have been added
Other suggestions:
- For the 'materials' selection, three samples were compared, the last is best but there is no reason why the last one is optimal to reproduce realistic radiolucency of the vertebrae.
R: An explanation has been added.
2. There are many linguistic errors and slips, as well as the reference errors via translating Words into PDF. Grammar and long sentences throughout the whole paper need to be checked and improved.
R: Errors and spelling have been improved
In brief, the value of the focused topic in this paper is high, but several aspects as mentioned, as well as the general readability, need to be addressed at first place.
Reviewer 3 Report
Courvoisier et al., reports a 3D printed models for AIS. They claim the benefits for testing and education. Unfortunately, due to errors left in text, the references and figures are missing. This made a difficulty for me to judge quality of the data and writing. There are also many typos and I also suggest English editing service. Examples: Line 30:Particularly published papers in the field of spine surgery dramatically grew from 4 in 2014 to 108 in 2021. Line 31: Different applications have been reported, if only educational purposes were sought out at first, now 3D printing for patient specific implants or surgical tools have been reported, for both adults and pediatric patients. Line 61: [ref Humbert &al. 2009]. Line 83: 2.23. D printing materials Line 85: Error! Reference source not found.): Table 1: far from the place where mentioned in text.
Author Response
The text was edited and improved.